# Biological Early Warning Systems: The Experience in the Gran Sasso-Sirente Aquifer

Federica Di Giacinto [1,*], Miriam Berti [1], Luigi Carbone [2], Riccardo Caprioli [1,3], Valentina Colaiuda [4,5], Annalina Lombardi [4], Barbara Tomassetti [4], Paolo Tuccella [5], Gianpaolo De Iuliis [6], Adelina Pietroleonardo [7], Mario Latini [8], Giuseppina Mascilongo [1], Ludovica Di Renzo [1], Nicola D'Alterio [1] and Nicola Ferri [1]

1 Istituto Zooprofilattico Sperimentale dell'Abruzzo e del Molise "G. Caporale" (IZSAM), 64100 Teramo, Italy; miriamberti2020@gmail.com (M.B.); riccardo.caprioli@arpalazio.gov.it (R.C.); g.mascilongo@izs.it (G.M.); l.direnzo@izs.it (L.D.R.); n.dalterio@izs.it (N.D.); n.ferri@izs.it (N.F.)
2 Officine Inovo, Engineering & Design Studio, 64100 Teramo, Italy; info@officineinovo.it
3 Agenzia Regionale Protezione Ambiente del Lazio—ARPA Lazio, 00187 Rome, Italy
4 CETEMPS, Centre of Excellence—University of L'Aquila, 67100 L'Aquila, Italy; valentina.colaiuda@univaq.it (V.C.); annalina.lombardi@aquila.infn.it (A.L.); barbara.tomassetti@aquila.infn.it (B.T.)
5 Department of Physical and Chemical Sciences, University of L'Aquila, 67100 L'Aquila, Italy; paolo.tuccella@aquila.infn.it
6 Ruzzo Reti Spa, 64100 Teramo, Italy; g.deiuliis@ruzzo.it
7 Consorzio di Bonifica Interno "Bacino Aterno e Sagittario", 67100 L'Aquila, Italy; info@cbaternosagittario.it
8 World Organization for Animal Health—OIE, 75017 Paris, France; m.latini@oie.int
* Correspondence: f.digiacinto@izs.it; Tel.: +39-0861-3321

**Abstract:** Biological early warning systems (BEWS) are installed worldwide for the continuous control of water intended for multiple uses. Sentinel aquatic organisms can alert us to contaminant presence through their physiological or behavioural alterations. The present study is aimed at sharing the experience acquired with water biomonitoring of the Gran Sasso-Sirente (GS-S) aquifer. It represents the major source of the Abruzzi region surface water, also intended as drinkable and for irrigation use. Besides the biomonitoring of drinkable water of the Teramo Province made by Daphnia Toximeter and irrigation water of the L'Aquila Province by Algae Toximeter, a novel sensor named "SmartShell" has been developed to register the behaviour of the "pea clam" *P. casertanum*, an autochthonous small bivalve living in the Nature 2000 site "Tirino River spring". The valve movements have been recorded directly on the field. Its behavioural rhythms have been analysed through spectral analyses, providing the basis for further investigations on their alterations as early warnings and allowing us to propose this autochthonous bivalve species as a novel sentinel organism for spring water.

**Keywords:** biological early warning system; Gran Sasso-Sirente aquifer; valvometry; behaviour; *Pisidium casertanum*

## 1. Introduction

Contaminants from industrial and urban sources can be accidentally or intentionally discharged at unpredictable times into the water. Common water monitoring programs are inadequate to ensure 24 h control, as they are organized by punctual samplings at pre-established periods of time [1]. Moreover, chemical analysis tools are not able to determine the concentrations of every compound existing in a water system due to time, cost and technical limitations [2]. Combined toxic effects, including synergetic and antagonistic ones, cannot be identified by chemical analysis tools. These limitations lead to the development of biological early warning systems (BEWS), which are based on the response of living sentinels (i.e., molluscs, algae, crustaceans, fish) to a contaminant or mixture of them [3]. BEWSs perform a real-time and continuous (24 h) monitoring of physiological and/or

behavioural parameters of organism alterations, potentially correlated to water pollution. Early warnings can be sent by SMS, e-mail, etc., to activate response actions [4].

The first research and technological advances on BEWS started in the early 1900s [5,6]. From initial trials based on the simple registration of the organism activities, it passed to the current advanced computational techniques such as video analysis [4–7].

Based on multiple endpoint parameters, numerous instrument models are available on the market [8]. Daphnia Toximeter (DTOX) and Fish Toximeter (®bbe moldaenke) analyse the swimming behaviour of *Daphnia magna* and fish, respectively. Mosselmonitor® (aquadect company) registers mollusc valve movements [9,10]. Easychem® Tox Early Warning (SYSTEA s.p.a.) and Toxcontrol® (microLAN) register bacteria bioluminescence alterations. Algae Toximeter (ATOX) (®bbe moldaenke) analyses photosynthetic activity inhibition of green microalgae. Moreover, many tools have been developed and applied by research institutes without the commercial scope, i.e., mussel actograph for shell gape monitoring [11], automated Grid Counter device for abnormal activity of *D. magna* [12], video-based movement analysis system to quantify behavioural stress responses of fish [13]. The Istituto Zooprofilattico Sperimentale dell'Abruzzo e del Molise "G. Caporale" (IZSAM) has developed a novel sensor named "SmartShell" for continuous monitoring of valve movements of marine and freshwater bivalves, capable of registering the movements of very small molluscs (height > 2 mm) impossible to allocate in other commercial tools [14].

Preferably, BEWSs are used in multiple panels covering different trophic levels, while the selection has to consider species-sensibility to contaminants according to pollution risk assessment of monitoring locations [15]. Belonging to decomposer trophic level, bacteria are very sensitive to a wide range of chemicals applicable to wastewater, surface water and groundwater [16,17]. Algae, as primary producers, are vulnerable to pesticides and metals [18] and are mainly used for surface water monitoring. Aquatic invertebrates, as primary consumers, are commonly used for ecotoxicological studies. The crustacean *Daphnia* represents the motile model for behavioural study: its swimming behaviour is recognised as a very sensitive biomarker in toxicity assessment [19,20]. Heavy metals, disinfection by-products, pesticides, etc., have been detected early in groundwater, surface water and wastewater through the water flea swimming alterations [19]. Bivalves, as filter feeder consumers with limited motility, occupy a prominent place in the passive and active biomonitoring [21]. They can accumulate contaminants and their valve movement behaviour is considered an important biomarker in ecotoxicology [22]. Mussels are mainly used for surface water monitoring by alerting the presence of organic and inorganic chemicals [21,22]. As fish are high on the aquatic food chain, they have been studied for decades considering many parameters (swimming, respiratory exchange, etc.) susceptible to low levels of contaminants in surface water and groundwater [8–23].

BEWSs have been installed worldwide, mostly after massive spills or in vulnerable locations. In China, during the Beijing Olympic Games, 40 BEWSs have been installed in water plants and in environmental monitoring stations in Beijing, Tianjin, Ji'nan, Guangzhou and Ningbo [4,24]. After the terrorist attack of 11 September 2001, the United States of America developed the Homeland Security Strategy applying BEWSs such as DTOX, ATOX, and fish monitor in drinking water security [25]. In Europe, biological monitoring was also encouraged by the Water Framework Directive [26] that transformed the concept of water quality monitoring, giving more importance to the ecological protection of water [27]. Nowadays, in the Rhine River, several physicochemical probes and BEWSs (DTOX, ATOX, mosselmonitor®) are working to rapidly detect pollution [28]. In Italy, one mosselmonitor® has been installed in the drinking water supply of Turin [29] and another one is present in Ferrara [30]. Currently, DTOXs continuously monitors drinkable water of Rome and Teramo.

Tap water of the Teramo province (Abruzzi region) is drained from the Gran Sasso (GS) aquifer, which is recognised as a vulnerable water source subjected to the Water Safety Plans at the national level [31,32], due to the underground infrastructures built inside [33]. In the 1980s, two highway tunnels and the Italian National Institute of Nuclear Physics

laboratory (INFN) were excavated in the core of the aquifer, posing a potential risk on its quality. In 2003, the presence of anomalous organic substances was found in its spring water [34]. Furthermore, on 9 May 2017, Teramo citizens were not authorized to drink potable water [35]. At the moment, an official multidisciplinary team is implementing the action plan according to the Water Safety Plans principles to ensure safe drinking water in Teramo city. The continuous monitoring of tap water is one of the planned key actions. Within this framework, DTOX was installed at the collection of potable water of Teramo, close to the INFN and motorway tunnel, as a vulnerable site to be controlled 24 h.

Water flowing from the Gran Sasso-Sirente (GS-S) aquifer is also exploited for irrigation use, since it constitutes the major part of regional surface water [36]. Pesticide pollution in aquatic ecosystems and in groundwater is a worldwide problem to be solved. A recent study assessed the ecological risk of pesticide mixtures in the alluvial aquifers in the Abruzzi region [37]. Continuous monitoring of irrigation water quality would avoid contamination and illegal use of banned substances that are yet persistent. Hence, IZSAM has started the monitoring of regional irrigation water through the BEWS "ATOX", highly sensitive to pesticides.

The GS-S aquifer feeds the Tirino River, which is considered one of the most important Italian "chalk-streams" [38]. Mainly constituted by groundwater, it exhibits less seasonal variation than other rivers, and it constitutes a very precious habitat. The first part of the Tirino River has been recognized as a Nature 2000 site (IT7110209). The surrounding area is subjected to agricultural and industrial activities, mostly in the urbanized area of Bussi sul Tirino and Popoli. Some water channellings have changed the natural assets of the river [39]. Considering the high environmental value, this water body should be under continuous control to avoid the loss of biodiversity. For this purpose, the novel sensor "SmartShell" presented in this study has been used for the first time to assess the behaviour of the pea clam (*Pisidium casertanum*), living in this area. It has been installed directly in the field, without any environmental modifications, which are necessary in the case of allochthonous indicator species use.

The main aim of this work is to report the experience on water biomonitoring of the GS-S aquifer, focusing on the development of a novel bio-sensor. A never-tested autochthonous bivalve has been proposed as a new sentinel organism for spring water. For further information, an overview of the monitoring results of the GS-S aquifer as a whole is briefly reported in the results section. Innovative technologies and methods internationally recognised as efficient tools have been used for water quality control, safeguarding human and ecosystem health.

## 2. Materials and Methods

### 2.1. Study Area: The Gran Sasso-Sirente Aquifer (GS-S)

The GS-S is situated in Central Italy and belongs to the Apennines ridge area, corresponding to part of the Gran Sasso–Laga National Park and part of the Sirente-Velino regional Park (Abruzzi region, Figure 1).

The aquifer system has an extent of about 700 km$^2$ [40], and the total drained discharge estimated on its springs is more than 18 m$^3$/s [41]. The hydrogeological system can be divided into two sub-units, corresponding to the GS aquifer, the wider the northern part, and the Sirente aquifer, in the south-western side of the Aterno River. From a structural-stratigraphic point of view, both units consist of fractured and karstified carbonate platforms [42].

According to Amoruso et al. [43], the GS is ~1000 m thick and retains $10^{10}$ m$^3$ of water. It is surrounded by well-defined aquitards, made by flysch deposits laying in the northern and western boundaries down to the Bussi town: no subterranean water exchange is present in those areas due to the tectonical overlapping with clayish layers alternated to sandstone. Fluvial and lacustrine deposits and debris layers with low permeability are mainly located on the south-western border of the GS aquifer and inside the massifs. With its very high permeability, the endoeric basin of Campo Imperatore, in the middle of the

main ridge, is the most important recharge area for the springs systems, mainly located in the northern and eastern boundary and characterized by quite constant discharge rates between 0.5 and 7 m$^3$/s.

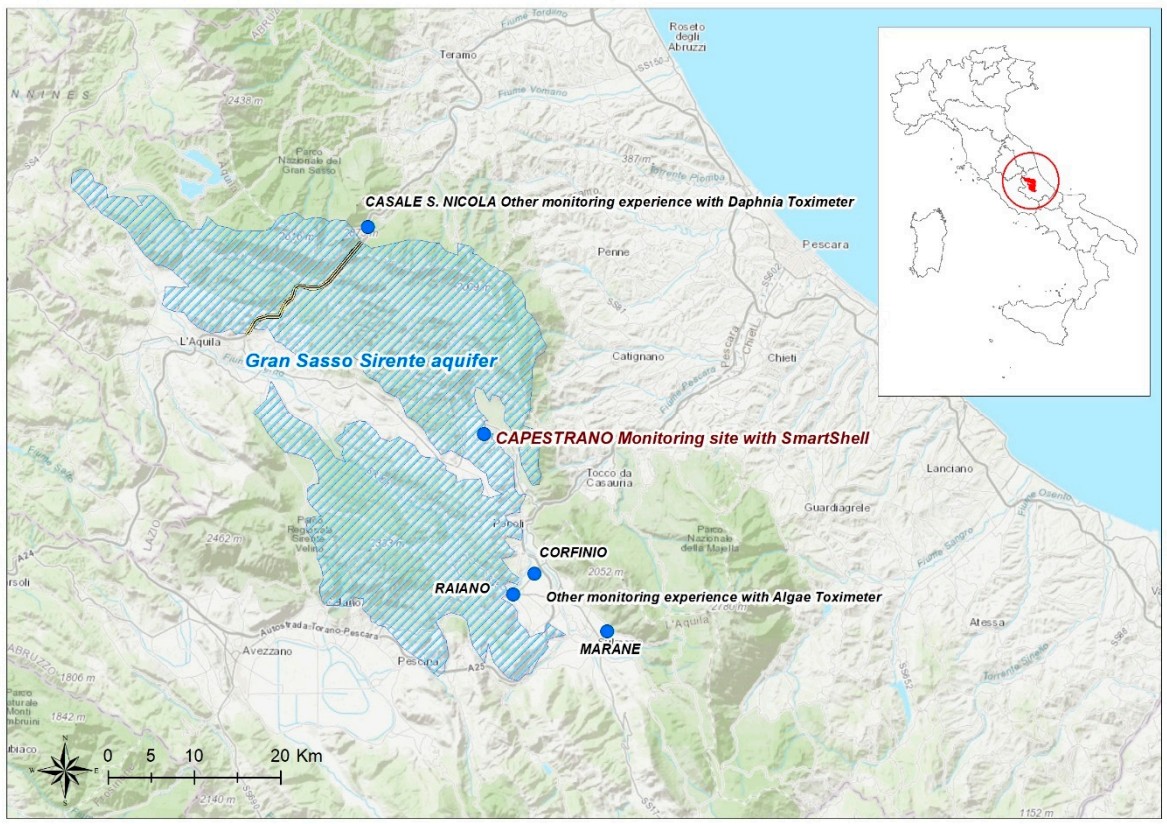

**Figure 1.** Map of the monitoring site in the GS-S aquifer where SmartShell has been installed and other water biomonitoring experience sites using BEWSs.

Boni et al. [44] have estimated a mean recharge rate in Campo Imperatore of about 700 mm/y, corresponding to the occurrence of an average precipitation of 1200 mm/y over the same basin [45]. The hydrogeological structure is NW-SE oriented, organized in a sequence of complex interconnected karstic reservoirs, frequently interrupted by low-permeable layers and faults that influence the underground circulation [46], resulting in progressively decreasing piezometric levels with a hydrostructural peak in the upper part [47]. Most of the subterranean drainage flows south-westward, feeding the Tavo, L'Aquila and Tirino springs, while a residual fraction is collected in the northern part, flowing into the Ruzzo springs. A hydrological subterranean exchange has been described between the GS carbonate massif and the Sulmona plain in the southern part [48].

The Gran Sasso mountain hosts the A24 motorway tunnel connecting the cities of L'Aquila and Teramo, as well as the INFN laboratories. The impressive infrastructure was built between 1969 and 1987 and consists of two 10 km-long tunnels placed at 970 m a.s.l, crossing the mountain core, and three 36,000 m$^3$ rooms, serving as INFN laboratories, that are connected by minor passages. The artificial structure intercepts carbonate rocks and the argillic-marly deposit and had a great impact on the massif hydrostructure during its building, causing the piezometric surface to vertically decrease about 600 m [47–49]. As a consequence, springs discharge significantly decreased, especially in Ruzzo springs. Well points have been installed along the tunnel path to drain water coming from the upper reservoir: a system of two 5 km-long channels subdivides the drainage water between L'Aquila and Teramo provinces, where it is exploited for drinking uses, for a total amount of about 1.5 m$^3$/s [42].

The GS aquifer consists of 7 groundwater bodies, whose characteristics are summarized in Supplementary Table S1.

The Sirente sub-system includes the southern part of the GS-S aquifer, respect to the Aterno river valley. Eastward, the Sirente unit is confined by the Sulmona plain, where relevant groundwater exchange with the fluvial and lacustrine deposits is present. The southern side of the aquifer is delimited by other overlapping carbonate deposits from the Marsicano massif, with no relevant groundwater exchanges, while the south-western limit is represented by the carbonate massif of Monte Pianeccia, where groundwater exchanges have been observed in the Celano plain. Finally, the northern boundary is characterized by water exchanges with the L'Aquila plain. The groundwater circulation scheme is described in Supplementary Table S2.

*2.2. Monitoring Site*

SmartShell has been mounted directly on the field, at the monitoring site (Figure 1) of GS-S water at Capestrano crayfish hatchery (Capestrano, L'Aquila, Itay). Here, the Tirino spring water flows directly in the hydraulic system of the hatchery. The Capestrano site is within Tirino springs which belong to the Colle Madonna groundwater catchment in the GS aquifer, but receive water also from other northern groundwater bodies up to the hydrostructural ridge in Gran Sasso. It is a group of springs that originates the namesake river, in three points: the Capo d'Acqua Lake (373 m a.s.l.), the Capestrano Lake (337 m a.l.m.) and Presciano (329 m a.l.m.). The left branch of the Tirino River arises from Capo d'Acqua, while the right branch results from the merging between Capestrano and Presciano.

The chemical/physical characteristics of the monitored water are listed in Table 1. The parameters have been registered with chemical/physical probe (AP-2000 model, aquaread®, Kent, UK).

**Table 1.** Characteristics of the monitored water.

| Monitoring Site | Water | BEWS Monitoring Period | Temp. (°C) | pH | Dissolved Oxygen (%) | Conductivity µS/cm |
|---|---|---|---|---|---|---|
| Capestrano | Tirino River spring | 30/03–12/04/18 – 05/07–27/07/18 | 12.30–13.50 | 6.64–7.14 | 89.7–87.4 | 668–726 |

Other water biomonitoring experiences of GS-S have been placed at (Figure 1):

- Casale San Nicola monitoring station (Isola del Gran Sasso, Teramo, Italy), where groundwater collected close to highway tunnels and INFN laboratory (technically named "Ruzzo sbarramento destro + sinistro") is continuously controlled by DTOX in the aqueduct network of the Teramo province (managed by Ruzzo Reti S.p.a.);
- Corfinio, Marane and Raiano channel and tanks (Sulmona, L'Aquila, Italy) in the irrigation network managed by Consorzio di Bonifica dell'Aterno Sagittario are monitored by ATOX.

*2.3. SmartShell in the Tirino River Springs*

SmartShell has been installed in March 2018 at the Capestrano crayfish hatchery (Capestrano, L'Aquila, Italy). It is a novel sensor developed by IZSAM, capable to measure the valve movements of bivalves, remotely and continuously, with high resolution (>0.1 mm). Differently from other systems, it does not require any attachments on the shells, thus reducing disturbance to the mollusc and allowing the use of small size bivalve species (height > 2 mm). The measurement principle is based on the detection of the distance between the proximity infrared (IR) sensor and mollusc shell. The principal components of the SmartShell are shown in Figure 2.

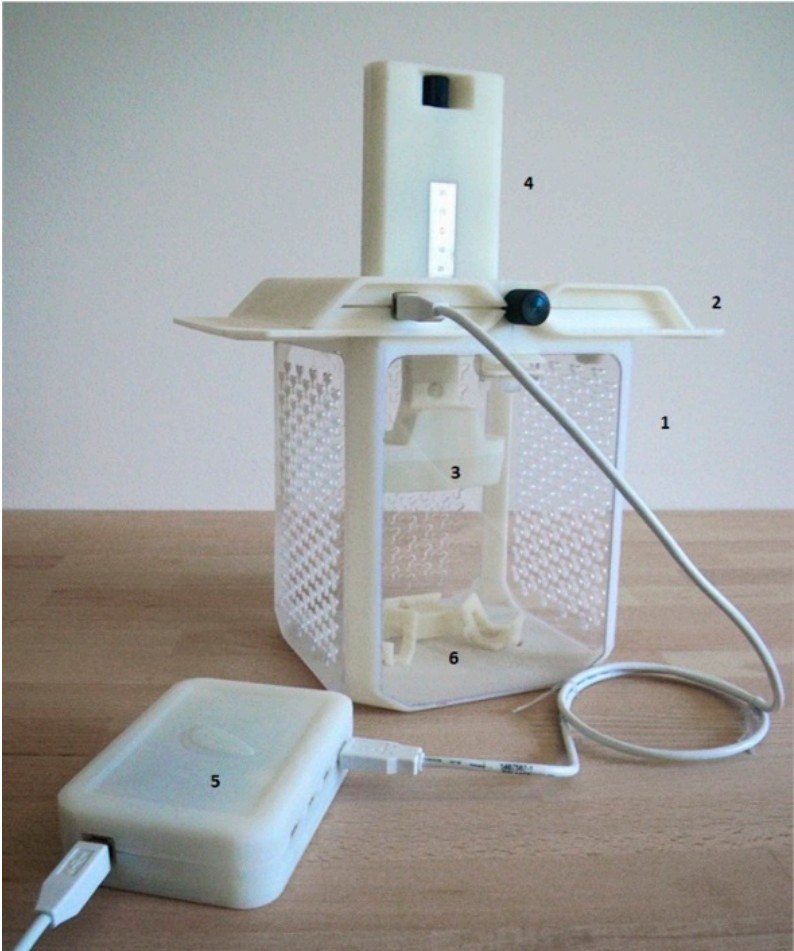

**Figure 2.** SmartShell components: 1—sensor tank; 2—sensor tank cap; 3—IR sensor containment chamber; 4—positioning system of sensor; 5—external case containing hub; 6—mollusc support.

It is based on the IR proximity sensor (model GP2Y0A41SK0F, sharp corporation, Japan) and an open-source hardware and software platform called "Arduino" (https://www.arduino.cc/ (accessed on 27 November 2020)) [50]. Each mollusc is glued on sensor support and IR light direction can be regulated on the shell by the positioning system of the sensor. The sensor detects the distance of the object and converts it into an analogue signal that is acquired by the hub acting as a data distribution node. The hub converts the signal into a digital one with a resolution of 10 bits and sends it to the excel software on a scale ranging from 0 to 1023 values. The IR sensor has a light emitting diode (emitter), a position-sensible photo detector—PSD (receiver) and a signal processing circuit. After reflecting on the mollusc shell, the emitted light beam hits the PSD, whose conductivity depends on beam fall position. The analogue distance value is proportionated to the outputted conductivity. The optical path of the reflected light beam is influenced by the different refractive indices ($n$) of the media it passes through. Emitted and reflected light beam of SmartShell sensor passes through air ($n$ = 1), polycarbonate ($n$ = 1.492) and water ($n$ = 1.333). This generates a difference between the measured distance and actual distance in water according to the description of Figure 3.

All plastic components of SmartShell are designed ad hoc and manufactured in polyamide using Subtractive Rapid Prototyping technology. Except for the cap, the sensor remains wholly immersed in the water during the test.

Valve movements are registered every 30 s, and a graph per mollusc is elaborated by the customized instrument software reporting the opening of the valves during the experimental time. An example has been reported in Figure 4.

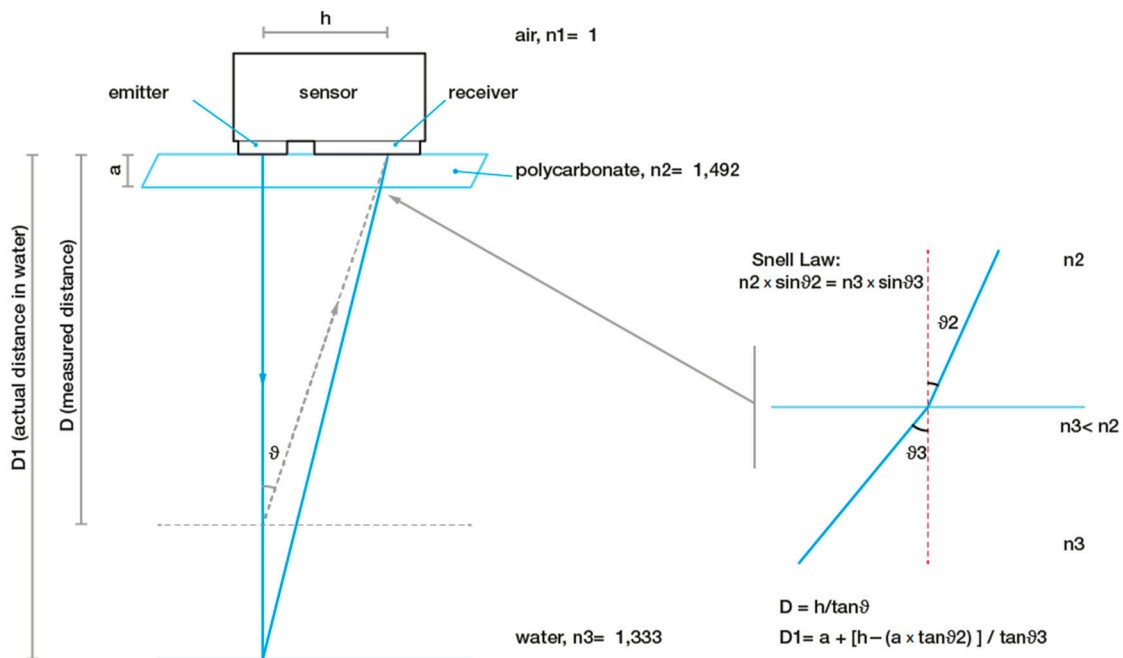

**Figure 3.** SmartShell: IR optical path and measurement of object distance.

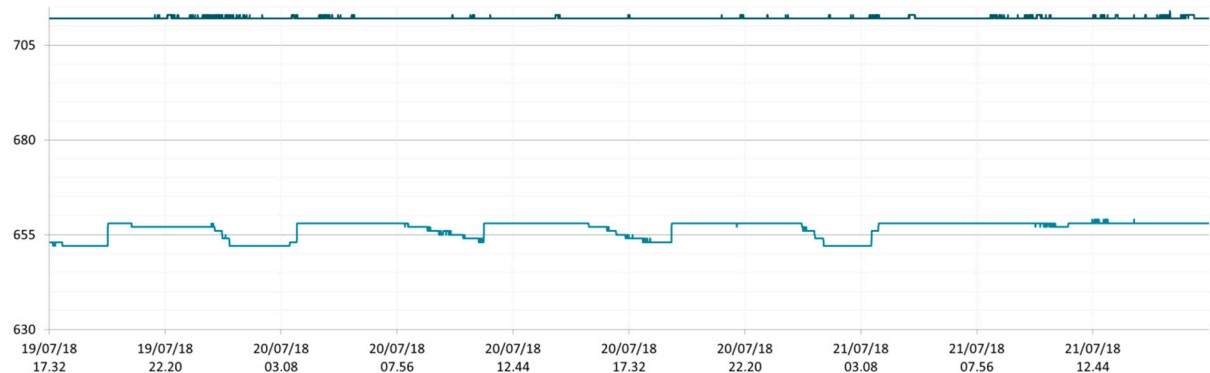

**Figure 4.** Graph continuously calculated by SmartShell software with the *X*-axis showing "time" and the *Y*-axis showing "valve opening". Each mollusc (named with code "HUBxx-xx") has a distinct pattern highlighted with specific colours, dark blue and light blue.

Further data analysis has established the maximum percentage of opening per specimen, elaborating the valve gape (VG). Five ranges of valve gapes (VG) have been established for the frequency calculation of results: VG $\leq$ 20%, 21–40%, 41–60%, 61–80% and $\geq$81% [51]. Time spent in each VG has been calculated by multiplying the number of observations in each VG type per registration time (30 s).

The sentinel organism chosen for the study was the "pea clam" *P. casertanum*, a small freshwater bivalve species autochthonous in the Tirino River. Study animals were collected near the monitoring site, measuring an average shell length equal to 3 mm. They were immediately observed by stereomicroscope (wild M3, Heerbrugg, Switzerland) to verify their vitality. The live specimens were first acclimatized in a tank with an open river water flow rate of almost 20 L/h, in IZSAM's crayfish hatchery at the Tirino River.

The taxonomic investigations were carried out in the laboratory with the use of morphometric analytical keys [52] using a scanning electron microscope—SEM (Zeiss DSM 940 A—Carl Zeiss Oberkochen, Germany) (Figure 5) [53] and through the biomolecular test.

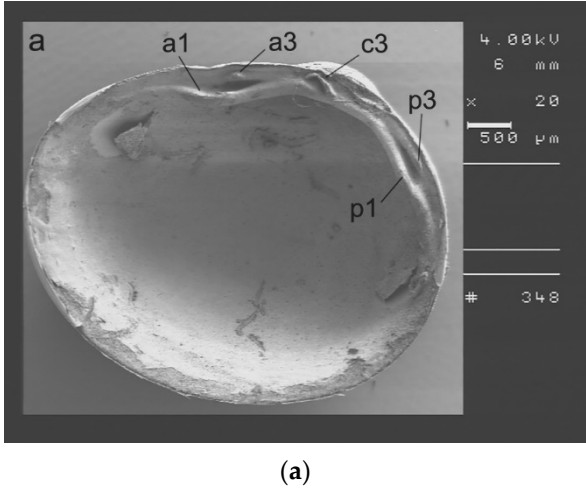 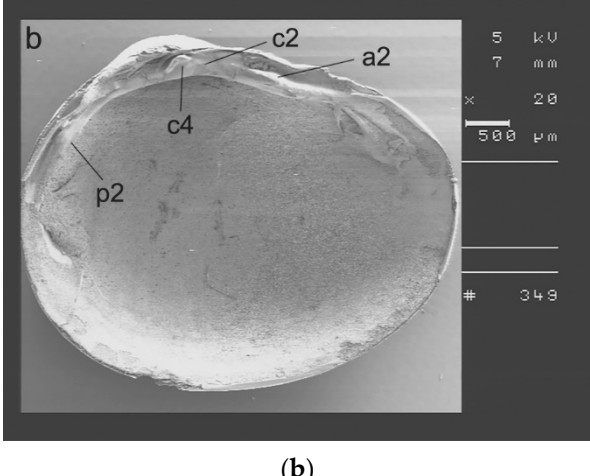

(**a**)                    (**b**)

**Figure 5.** Morphometrics analyses of valves of *P. casertanum*. (**a**) Insight view of right valve with cardinal tooth c3, anterior lateral teeth a1 and a3, posterior lateral teeth p1 and p3. (**b**) Insight view of left valve with cardinal teeth c2 and c4, anterior lateral tooth a2, posterior lateral tooth p2.

Total DNA was extracted from soft tissue samples by Maxwell® 16 Tissue DNA Purification Kit (Promega, Madison, WI, USA) and tested by conventional PCR with specific primers [54] amplifying a portion of 700 bp of the Cytochrome Oxidase I (COI) gene. All PCR products were sequenced using the Sanger method. The consensus sequences were obtained using Unipro UGENE v.32 (Okonechnikov K, Golosova O, Fursov M, the UGENE team. Unipro UGENE: a unified bioinformatics toolkit. Bioinformatics 2012 28: 1166–1167. doi:10.1093/bioinformatics/bts091) and BLASTn analysis was performed to identify the closest publicly available sequence in the Genbank database. The highest nucleotide identity (100%) was evidenced with *Pisidium casertanum*, already registered as autochthonous species in the Tirino River [55].

After acclimatization of 15–20 days, molluscs were inserted on the support of SmartShell (Figure 6). Two groups of 2 individuals were observed directly on the field, respectively, for 12 and 23 days, by putting them in the same tank of the acclimatization with a continuous river flow rate.

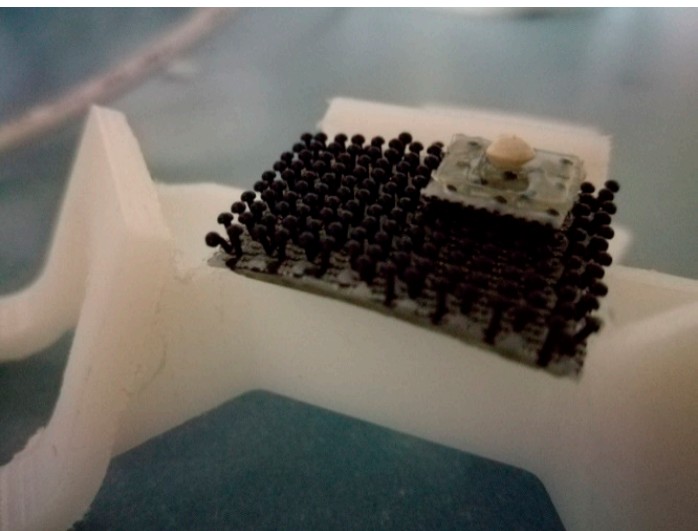

**Figure 6.** One specimen of *P. casertanum* installed on the support of SmartShell.

### 2.4. Spectral Analysis of Valve Movements

In order to investigate dominant frequencies of molluscs' valve behaviour, the Fourier transform has been applied to opening time series of molluscs HUB03-01 of the first test and HUB03-02 of the second test [4,56]. The transform decomposes a signal function represented in a time domain (i.e., the opening time-series) into its characteristic sinusoidal components in the complex field, characterized by their own amplitudes, phases and frequencies. The absolute value of the transform is the original frequency value of the original time-series, while the complex argument represents the phase offset of predominant sinusoidal functions that compose the signal. The discrete Fourier transform $Y(k)$ for a given time-series of $n$ samplings is derived by the following formula:

$$Y(k) = \sum_{j=1}^{j=n} X(j) W_n^{(j-1)(k-1)},$$ (1)

where $k = 0, \dots, n-1$ and function $W$ is given by:

$$W_n = e^{(-2\pi i)/n}.$$ (2)

Before carrying out the spectral analysis, the raw signal of the opening of both molluscs has been filtered, in order to remove spike values able to alter results. Primarily, the first two days of sampling were removed from the two time-series, as they correspond to an acclimatization period for the *P. casertanum* due to the instrument installation that inevitably perturbed surrounding environmental conditions. Subsequently, spike opening values lower than 701 and 652 were removed from mollusc HUB03-01 and mollusc HUB03-02 measurements, respectively. Those values can be considered unreliable opening data because they are under the minimum value appointed during the positioning closed mollusc.

Finally, the resulting signal has been smoothed using a moving average filter with a span of 240 samplings. Before the application of the Fourier transform, the presence of a trend in both molluscs opening time-series has been verified through a regression analysis, consequently, data have been detrended using the ordinary last squares method as described in Warner [57].

### 2.5. Other Biomonitoring Experiences in GS-S: DTOX in Drinkable Water of Teramo Province and ATOX for Control of Irrigation Water of L'Aquila

DTOX has been installed in April 2019 at Casale San Nicola monitoring station (Isola del Gran Sasso, Teramo, Italy). It is a BEWS able to detect toxic substances in the water through the registration of water flea behaviour anomalies. The measurement principle is based on a statistical analysis of swimming trajectories of 20 crustaceans (*Daphnia magna*) in a flow of water sample of 2–3 L/h, passing through the two measurement chambers [58,59]. The scheme of the principal components and the analytical procedures are available at https://www.bbe-moldaenke.de (accessed on 27 November 2020). The results of the monitoring programme (24 h) with DTOX, the weekly Toxic Index, are presented in this article from 4 January 2020 until 31 March.

ATOX has been installed in April 2019 at IZSAM laboratory of Teramo city. It is a BEWS enable to detect toxic substances in the water through the registration of green microalgae photosynthetic activity inhibition. The measurement principle is based on the determination of the fluorescence spectrum and fluorescence kinetics of algae [60]. The scheme of the principal components and the analytical procedures are available at https://www.bbe-moldaenke.de (accessed on 27 November 2020).

Surface water has been sampled at the irrigation net managed by the Consorzio di Bonifica del Bacino dell'Aterno—Sagittario. Three hundred litres of water were collected from each sampling site and immediately transported in a tank to the lab. The duration of each analysis was almost 7 days. Data on % inhibition have been analysed.

## 3. Results

### 3.1. SmartShell in Tirino River Spring

Valve movements of four specimens belonging to *P. casertanum* have been observed directly on the field at the crayfish hatchery of Capestrano [AQ]. Two individuals have been simultaneously analysed from 30 March 2018 to 12 April 2018 (first study group). The other two were examined from 5 July to 28 July 2018 (second study group). The first group showed dissimilar behaviours: one individual (HUB03-01) registered a flapping graph with the alternation of activity and inactivity cycles of valve movements. The other one (HUB03-02) had the valves almost always opened, without any flapping activity (Figure 7). At the end of the test, the mollusc was alive and was incubating seven juveniles ready to be released (Figure 8).

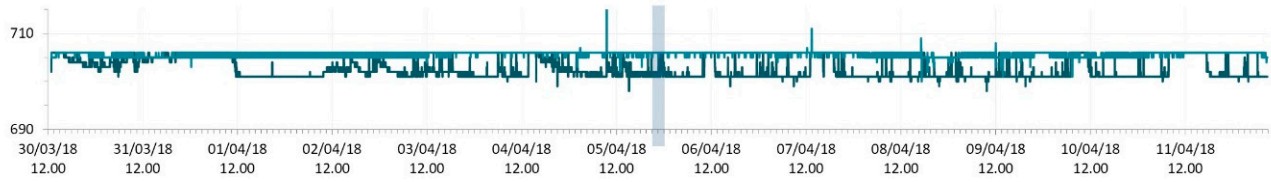

**Figure 7.** Raw patterns of valve openings of two individuals of *P. casertanum* included in the first study group, from 30 March 2018 to 12 April 2018: HUB03-01 in blue colour, HUB03-02 in heavenly colour.

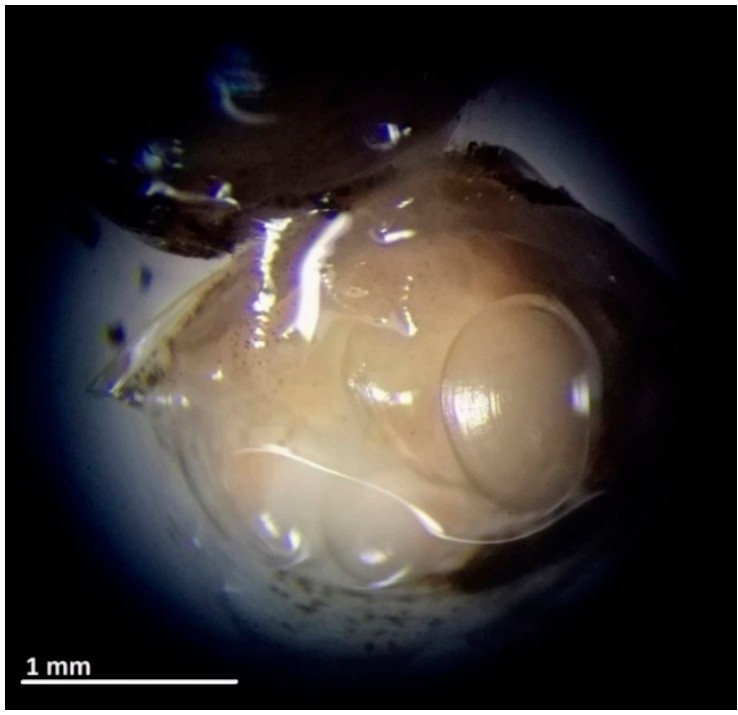

**Figure 8.** Mollusc HUB03-02: adult incubating 7 juveniles ready to be released (stereomicroscope 40× magnification).

In the second group of molluscs, the HUB03-01 mollusc (dark blue in Figure 9) was almost always open with frenetic and continuous valve movements for the first 10 days, then the specimen died (Figure 9). The other bivalve HUB03-02, highlighted with the light blue colour had clearly distinguishable periods of activity and inactivity, repeated within 24 h. Such periods did not seem to be related to the photoperiod. This mollusc was alive at the end of the test.

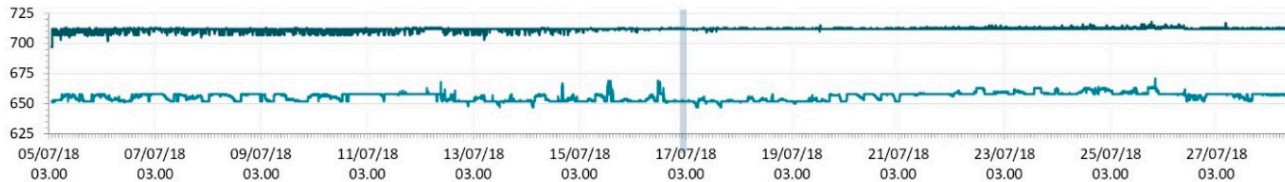

**Figure 9.** Raw patterns of valve openings of the two individuals of *P. casertanum* included in the second study group, from 5 to 28 July 2018: HUB03-01 in blue colour, HUB03-02 in heavenly colour.

The patterns of the two molluscs included in the second study group are better detailed in Figures 10 and 11. The HUB03-01 was always open with a continuous flapping from 10 to 12 July (Figure 10) until its death (Figure 11). The HUB03-02 (light blue in the graph), presented an initial trend more irregular in its rhythms (Figure 10), with long intervals of activity interspersed with short periods of inactivity. This behaviour became more regular during the following days, with three or four activity/inactivity cycles per day (Figure 11).

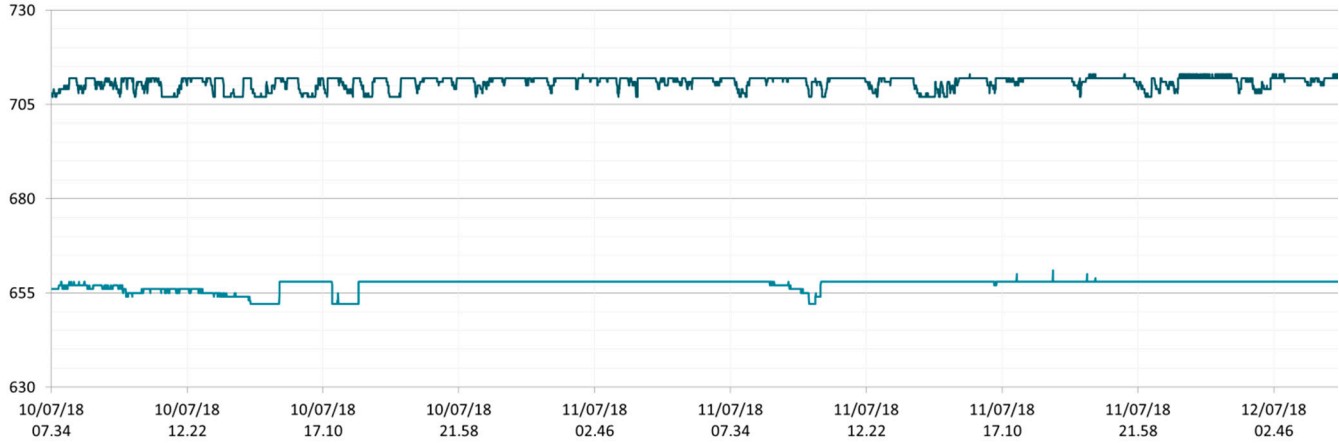

**Figure 10.** Zoom of the raw patterns of valve openings of two individuals of *P. casertanum* included in the second study group, from 10 to 12 July 2018: HUB03-01 in blue colour, HUB03-02 in heavenly colour.

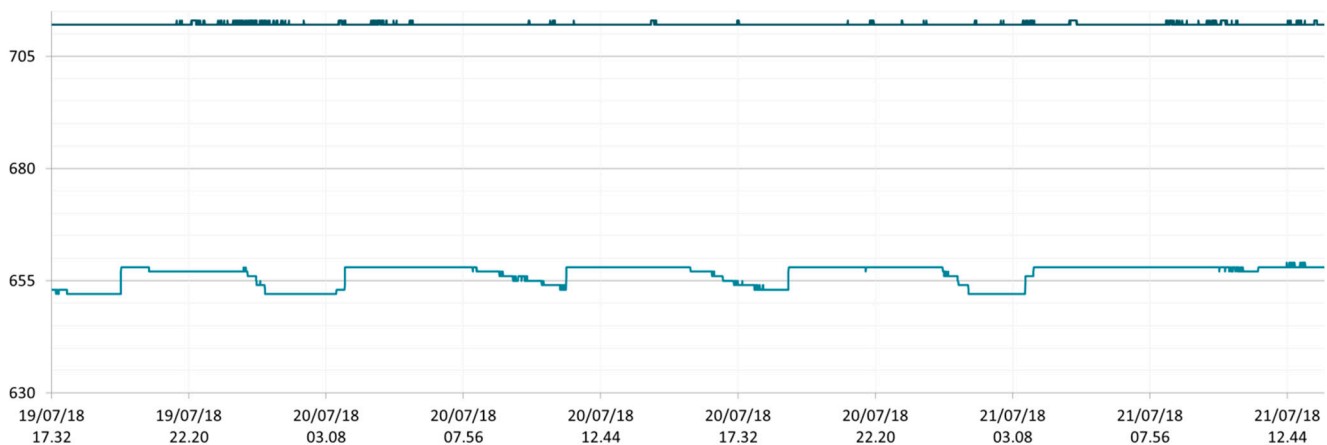

**Figure 11.** Zoom of the raw patterns of valve openings of two individuals of *P. casertanum* included in the second study group, from 19 to 21 July 2018: HUB03-01 in blue colour, HUB03-02 in heavenly colour.

The mollusc behaviour was further investigated by evaluating the extent of valve opening during the study period. Figures 12 and 13 show the average percentage of time spent by molluscs with given VG ranges.

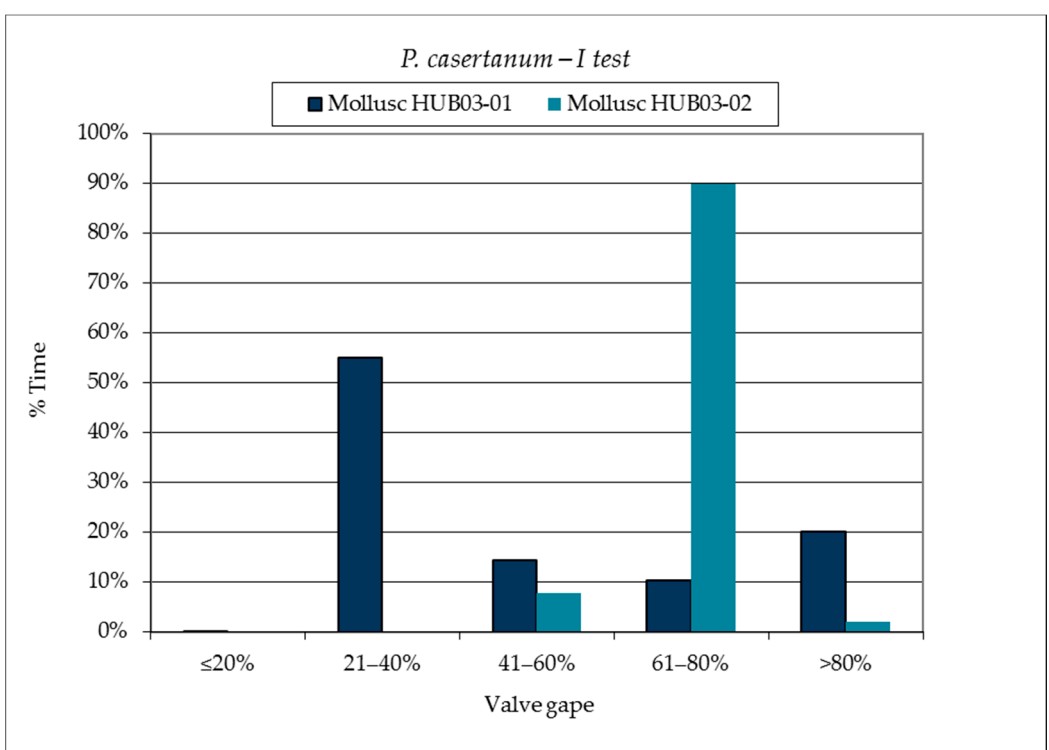

**Figure 12.** Average percentage of time spent in established valve gape ranges by molluscs in the first study group, from 30 March 2018 to 12 April 2018. Valve gapes is reported as % of valve openings.

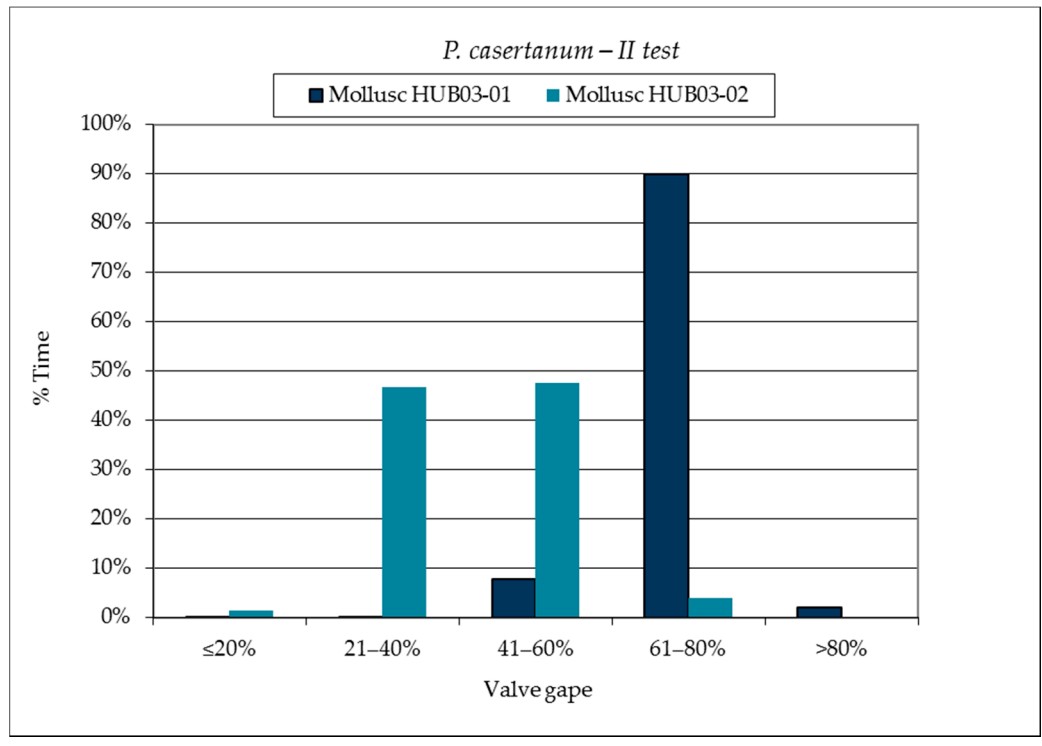

**Figure 13.** Average percentage of time spent in established valve gape ranges by molluscs in the second study group, from 5 to 28 July 2018. Valve gapes is reported as % of valve openings.

In the first study group (Figure 12), mollusc HUB03-01 showed a flapping behaviour, with about 50% of the time spent in the VG category of 21–40%. Conversely, mollusc HUB03-02 was always fixed at VG equal to 61–80% without any valve movements (Figure 12).

In the second study group, the mollusc HUB03-01 spent 90% of the time at VG of 61–80%, the other HUB03-02 ranged equally between the two VG categories 21–40% and 41–60% (Figure 13).

Fourier transform analysis of valve movements has been applied to the mollusc showing a flapping behaviour: HUB03-01 of the first study group, from 30 March 2018 to 12 April 2018, and to HUB03-02 of the second study group, from 5 to 28 July 2018. As shown in Figure 14, the graph of the dominant period resulting from spectral analysis of the mollusc HUB03-01 detected ~4 daily dominant periods and 2 periods slightly exceeding 1 day. The second mollusc, HUB03-02, showed dominant periods in ~5–6 days, and ~4 activity periods per day (Figure 15). In Table 2, further detail on the first six dominant periods for both molluscs is given.

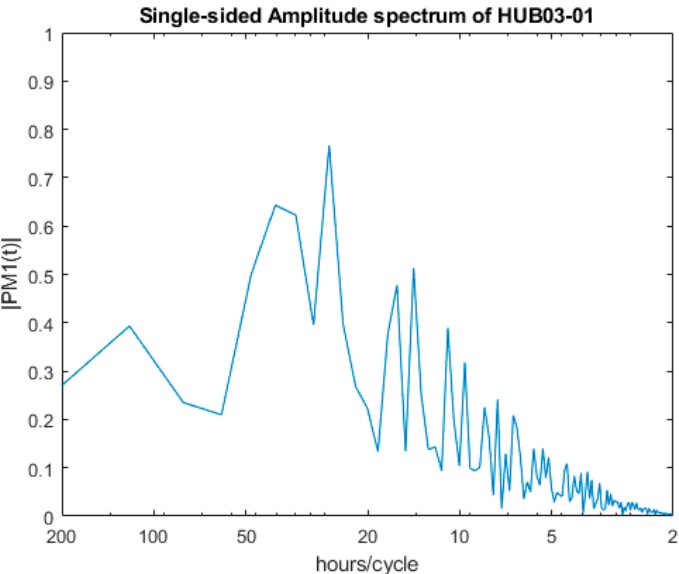

**Figure 14.** Fourier transform analysis of valve openings of mollusc HUB03-01 in the first study group, from 30 March 2018 to 12 April 2018 (x-axis in semi-logarithmic scale).

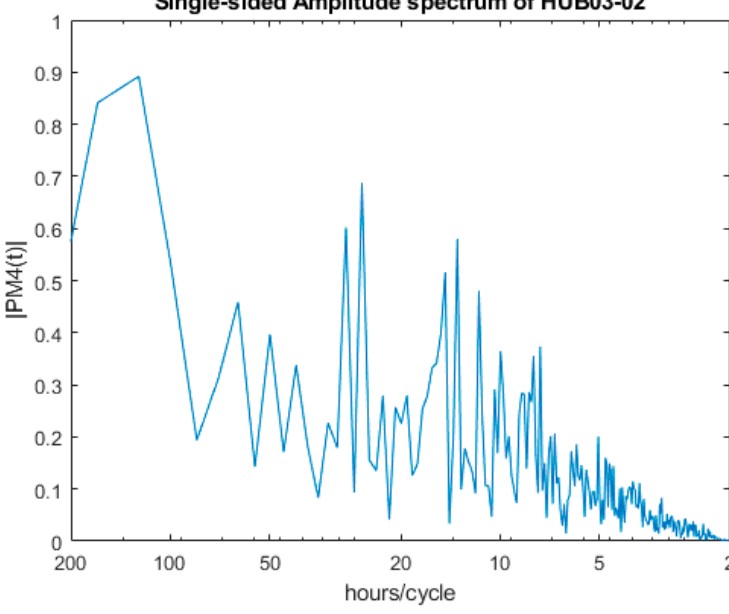

**Figure 15.** Fourier transform analysis of valve openings of mollusc HUB03-02 in the second study group, from 5 to 28 July 2018 (x-axis in semi-logarithmic scale).

**Table 2.** Dominant periods (hours/cycles) in descending order from the 1st to the 6th for HUB03-01 and HUB03-02.

| Dominant Period | HUB03-01 | HUB03-02 |
|---|---|---|
| 1st | 25.70 | 124.00 |
| 2nd | 40.04 | 166.03 |
| 3rd | 34.33 | 26.20 |
| 4th | 14.14 | 29.29 |
| 5th | 16.02 | 13.46 |
| 6th | 10.92 | 14.65 |

*3.2. DTOX in Drinkable Water of Teramo Province and ATOX for Control of Irrigation Water of L'Aquila*

Biomonitoring of drinkable water of Teramo province from 4 January 2020 until 31 March 2020 was performed continuously performed by DTOX (24 h) without any launched alarms. Examples of the recorded weekly toxic index are reported in Figure 16.

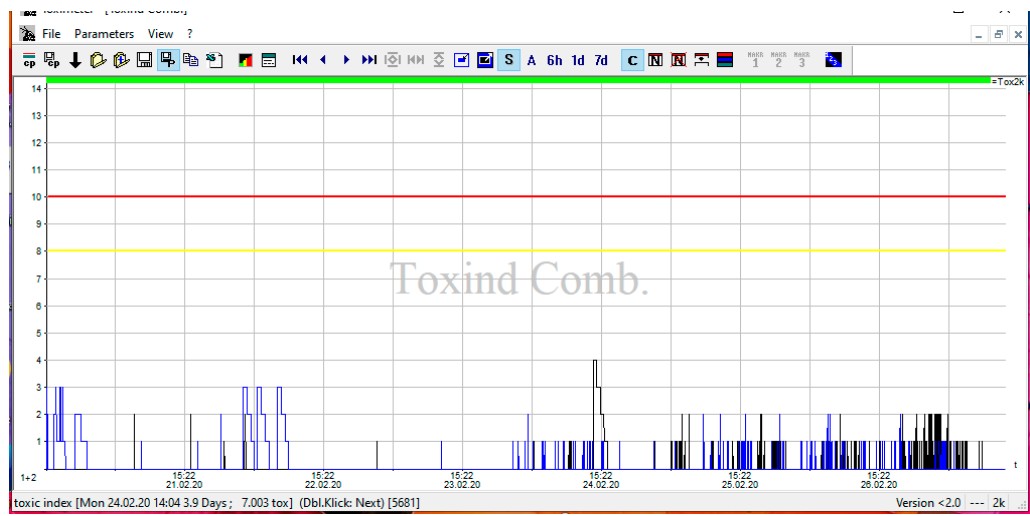

**Figure 16.** Toxic index combined of analysis from 20 to 27 February 2020.

All surface samples collected from the irrigation net showed almost 0% of inhibition in ATOX (Table 3). Accordingly, no alarms were launched.

**Table 3.** ATOX monitoring data.

| Analysing Date | Location | Maximum % Inhibition | Minimum % Inhibition |
|---|---|---|---|
| 23–26 June 20 | Marane | 0.34 | −0.94 |
| 20–27 July 20 | Corfinio | 1.40 | −0.96 |
| 27–31 July 20 | Raiano | 0.12 | −0.96 |

## 4. Discussion

In bivalves, the behaviour of valve movement is correlated to several activities, i.e., filtration, feeding, locomotion, reproduction, etc. For this reason, it has been investigated for a long time, also through the development of different kinds of registering "valvometers" useful for ethological and ecotoxicological studies [51]. Behavioural sublethal effects reflect the complex system of the organism's response to disturbances. The combination of physiological and environmental factors causes a reaction that could represent a set of acute cumulative effects, not detectable by classical methods. They occur precociously and at very low contaminant concentrations. Hence, the behavioural endpoint is very useful to evaluate how toxicants exert their effects and for early warning signalling in BEWS.

To date, some bivalve species such as *P. casertanum* have never been allocated in a "valvometer" due to their small dimensions. Indeed, the available commercial tools are based on sensors too big to be glued on the valves of this small mollusc. Moreover, such sensors do not have the sensibility needed to detect their little movements.

The SmartShell tool developed by IZSAM is able to register at distance the behaviour of molluscs with shell height >2 mm. This system is still unpublished in the scientific literature, as only preliminary results concerning laboratory testing with eight species of marine and freshwater mollusc species and reference toxicant have been described in a poster during EUROTOX 2019 [14]. This tool can be used both as a "valvometer" and as an "early warning system" with the appropriate algorithm.

The basic idea in SmartShell is that each bivalve species has a very characteristic rhythm of valve opening. Any deviation from the basic pattern should be considered as an early signal of water disturbances. Starting from this hypothesis, the initial step is to analyse in detail the normal behaviour of each bivalve species in order to elaborate a specific early warning algorithm for each one.

In this study, the behaviour of *P. casertanum* has been recorded for the first time, showing very particular characteristics. From the spectral analysis of valve movements data, the normal rhythms of activity and inactivity period were repeated about four times per day, and another one is registered on the fifth day. A circadian rhythm has been registered one time a day in *Mytilus galloprovincialis* [56] and in *Anodonta anatina* [61]. Conversely, multiple rhythms have been observed in *Anodonta cygnea* [62], *Unio pictorum* [63], in *Corbicula fluminea* [64]. Sometimes these rhythms are correlated to diurnal/nocturnal cycles, in other cases to feeding behaviour or tidal frequency [65]. Moreover, examining valve movements for a long period of time is useful to comprehend the effect of natural environmental changes and natural conditions (temperature, food, etc.) [66,67].

In this study, the behaviour of an adult mollusc (HUB 03-02 of the first study group) ready to release seven juveniles was also recorded. *Pisidium* species, in fact, are hermaphrodites incubating their progeny until release [53]. This mollusc was almost fixed at VG ranging from 61 to 80%. It did not show any rhythmical pattern. This demonstrates that behaviour of viviparous bivalves is greatly influenced by the reproductive period. Before juveniles release, the mollusc did not move valves for 12 days at least. Two *P. casertanum* individuals showed valve gapes ranging from 21 to 40% for most of the observation period. This behaviour is quite unusual since studies conducted using "valvometers" on other bivalve species showed that their valves were are almost completely opened for most of the time [9,68]. Instead, it was almost confirmed in the study on the normoxic and anoxic heat output of the freshwater bivalves *Pisidium amnicum* and *Sphaerium corneum* [69]. They registered regular periods of behavioural and metabolic quiescence in normoxic conditions through the use of heat-flow microcalorimeter. A normoxic rhythm of heat output was registered with short bursts of high heat output of almost 1.5 h alternated with longer low heat flux suggesting a period of shell closure of 9 h. Therefore, the observed behaviour detected a long quiescence period surely almost three times per day. Metabolic quiescence in aquatic invertebrates is usual in the unfavourable environmental conditions such as lack of food, extreme temperature and anoxia.

For *P. casertanum,* this first study should be extended by investigating the behaviour pattern of this species during different seasonal periods, including the reproductive ones. Further studies are also needed to confirm the observations on the extent and the rhythms of valve gapes, and to identify potential interfering causes, with the aim of developing a reliable algorithm for early warning signalling. Being a cosmopolitan species, *P. casertanum* is easy to find in the environment, and, being an autochthonous species, its use is allowed in Natura 2000 sites like in the Tirino springs according to Habitat Directive [70]. Living in oligotrophic cold waters, this bivalve represents the best sentinel for this kind of water. Using the combination SmartShell with *P. casertanum*, the continuous monitoring of river springs could be arranged without any environmental modifications.

Concerning the other biomonitoring experiences carried out in the GS-S aquifer, DTOX has been chosen for continuous control of tap water, since *D. magna* is one of the most sensitive species for a wide range of compounds. ATOX has been selected for irrigation water, being the most reliable for pesticide detection. They did not launch any kind of early warnings by representing a good practical experience for safeguarding water bodies intended for multiple uses.

**5. Conclusions**

This study reported the experience gained through the use of different BEWS in different sensitive points of the waters of the GS-S aquifer, choosing the most appropriate tool and sentinel for each site. A never tested autochthonous bivalve, *P. casertanum*, has been studied trough SmartShell. Being immotile and filter-feeding, this autochthonous mollusc represents a good sentinel candidate to estimate natural environments as spring water through BEWS.

Our results may represent the basis to establish stable multiple trophic BEWSs for the control of the GS-S aquifer in specific sites, to be appointed according to the European and Italian Legislations concerning the Water Security Plan [71] and the protection zones (Italian National Decree 152/2006) [72]. BEWSs exploiting innovative technology are essential to ensure water protection, which also means to guarantee human and ecosystem health.

**Supplementary Materials:** The following are available online at https://www.mdpi.com/article/10.3390/w13111529/s1, Table S1: Classification of the Gran Sasso aquifer groundwater bodies, main hydrological characteristics and groundwater circulation, Table S2: Classification of the Sirente aquifer groundwater bodies, main hydrological characteristics and groundwater circulations.

**Author Contributions:** Conceptualization, F.D.G.; methodology, F.D.G. and M.B.; software, L.C.; formal analysis, R.C., G.M., L.D.R.; data curation, V.C. and P.T.; writing—original draft preparation, F.D.G. and V.C.; writing—review and editing, R.C., M.B., A.L., B.T.; visualization, G.D.I., A.P., M.L.; supervision, N.F., M.B., N.D.; project administration, N.F.; funding acquisition, N.F. All authors have read and agreed to the published version of the manuscript.

**Funding:** This research was partially funded by REGIONE ABRUZZO—Legge Regionale 27 agosto 1982 n. 59 "Controllo della salubrità delle carni ittiche"—year 2017.

**Institutional Review Board Statement:** Not applicable.

**Informed Consent Statement:** Not applicable.

**Data Availability Statement:** Data are contained within the article and supplementary materials. More detailed data may be requested from authors.

**Acknowledgments:** We would like to express special thanks to Sandro Pelini and Paola Di Giuseppe for map and graphics, Annarita D'Angelo for SEM analysis, Maurilia Marcacci and Valentina Curini for PCR, Gianfranco Diletti and Roberta Ceci for their technical supervision.

**Conflicts of Interest:** The authors declare no conflict of interest. The funders had no role in the design of the study; in the collection, analyses, or interpretation of data; in the writing of the manuscript, or in the decision to publish the results.

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
