# Peer review of "Biological Early Warning Systems: The Experience in the Gran Sasso-Sirente Aquifer"

_water, doi:10.3390/w13111529_

Round 1

Reviewer 1 Report

The paper have a very excellent idea and good present about the water biological early warning application. The paper is very organized, and the language is well written. I think it can give us a very  example for water biomonitoring applications .

Reviewer 2 Report

Review of: “Biological early warning systems: the experience in the Gran Sasso - Sirente aquifer”

The manuscript is overall well written with a good interpretation of the results. Presented data are most probably of good quality, but the international context of the work is probably insufficiently justified. However, considering the grooving importance of clean water and novelty of research my suggestion is to accept the article for publication.

This manuscript is a resubmission of an earlier submission. The following is a list of the peer review reports and author responses from that submission.

Round 1

Reviewer 1 Report

In this work, the authors attempt to report a case study of water biomonitoring of an aquifer executed with different Biological Early Warning Systems (BEWS) in different vulnerable points in this region. Overall, the topic is quite interesting as a lot of efforts have been dedicated lately to the field of BEWS. Moreover, the fact that the authors have developed a new system based on the valve movements of Pisidium casertanum highlights the importance of having different systems based on different organisms from the trophic chain. However, the manuscript if quite poorly written without a clear structure and is difficult to read or interpret results, which are of questionable quality given the method used (no positive control well developed at the lab scale before transferring the systems to the field) and the chaotic description of the different BEWS used. If the novelty of this study is the development of the system based in the valve movements of Pisidium casertanum, the authors should just focus in this and cut all the part related to the other systems.

The presentation of the results (i.e. 32 figures in total!) is not clear and does not help the reader get an impression of the overall results: These seem to simply be put together quickly with little thought of helping the reader. The authors may consider the use of the supplementary material in a wisely manner where they could put a lot of information (mainly from the M&M section) which is now presented in the main manuscript. Finally, the clear statement of the objectives of the study is missing in the abstract section, thus it is not easy to grasp the purpose of the work after reading it. Also, there is no a conclusion section at all. Therefore, the overall presentation of the manuscript and execution are neither good enough or up to par with publication standards and suggest to reject the article. I have already made some suggestions to the authors for places that they could fix the presentation of results, which may help for future versions of the manuscript to be published.

Author Response

Dear Referee,

many thanks for your comments and amendments. They gave us the opportunity to improve the manuscript. Please find enclosed our reply to your report.

Kind Regards

Reviewer 2 Report

This work investigates the application of a new sensor (Smartshell), which registers the valve movement of the pea clam, an aquatic sentinel which have never been explored before. The research topic is of paramount importance and helpful for the protection of water environments, guaranteeing human and ecosystem health. The experiments were well programmed and the paper is very well organized and written. Consequently, I strongly recommend its publication in Water.

Author Response

Dear Referee,

many thanks for the words. We are very pleased and honored to receive so much interest in our work! It gives us further strength to progress beyond.

Kind Regards

Reviewer 3 Report

This is an interesting manuscript that reports on the use of continuous biomonitors for as an early warning approach for assessing water quality in the Gran Sasso-Sirente aquifer, Italy. Three types of organisms are trialled in this study: Daphnia magna, algae and a species of freshwater clam.  

I am concerned about the novelty of some parts of this manuscript.  The Daphnia Toximeter (DTOX) and bbe® Algae toximeter (ATOX) are commercial products marketed by bbe Moldaenke, Germany.  Details about these 2 products that are presented in the manuscript are also provided online by this company (https://www.bbe-moldaenke.de/en/products/overview.html).  Therefore, the use of these devices is not novel and there is no need to provide so much detail on them in the methods.  The three BEWS were used in different locations, so it was not possible to provide any comparison of their performance including sensitivities.  Therefore, the data presented using the DTOX and ATOX is not novel.

As the authors point out (line 707), the real novelty in this manuscript is the clam sensor.  I would recommend that this paper only focuses on this clam sensor.  This will also condense the length of the manuscript. I therefore recommend that this paper is reconsidered after major revision.

Minor comments

General use capitals for the name of places.  E.g. Tirono River, Suelmona Plain, Marane Channel, Capo d'Acqua Lake

L69/71 Algae, as primary producers

delete pesticides or herbicides, as herbicides are pesticides.

What do you menan that algae are used for surface water control?

L 75 wastewater is one work

L 82 worldwide

L 89 physicochemical

L 96 in the 1980s

L111 The GS-S aquifer feeds the Tiron River

L26 autochthonous

L 133 The GS-S

L238 able not enable

L324 100% survival pf daphids

L311 was tested

L327 Daphnia magna

L332/333 Replace 'has been started' with 'are presented'

L350 Chorella vulgaris in italics

L351 algae

L354 algal concentration

L456 delete 'fixed'

L358, 366, 428 Be consistent on using the abbreviation for litre

The results of the reference results can be verbally report in the results and the graphs can be presented in a supplementary file.

L705 delete 'subsequently'

L706/707 delete 'different'

L721 what do you mean that they occur precociously?

L736 Apart from the death of one individual, the animals..

Author Response

Dear Referee,

many thanks for your comments and amendments that are very important to improve the manuscript.

Please find enclosed our response in the journal template.

Kind regards

Round 2

Reviewer 3 Report

Congratulations to the authors for greatly improving this version of the manuscript.  In my opinion, only minor edits are required for this manuscript to the ready for publication.

Line 19 installed worldwide

L23 replace 'with unxplored' with 'using a'

L68 contaminants

L71 producers are vulnerable

L79/80 alerting the presence of organic

L80 As fish are high on the aquatic food chain, they have..

L84 BEWs were installed

L86/87 After the terrorist

L90 in the Rhine

L91 rapidly detect pollution

L102 safe drinking

L122 time to assess the behaviour

L374/5 unreliable

L394 12th

L416 and 431. The print in Figures 9 and 11 are unclear and difficult to read.

L434 until it died

L509 on shells sensors too big for

Author Response

Dear Reviewer,

many thanks for your comments and amendments that are improved our manuscript. Please find enclosed our response in the attachment. 

Kind Regards

Authors
